# Technical note: $CO_2$ is not like $CH_4$ – limits of and corrections to the headspace method to analyse $pCO_2$ in freshwater

Matthias Koschorreck[1], Yves T. Prairie[2], Jihyeon Kim[2], Rafael Marcé[3,4]

[1]Department Lake Research, Helmholtz Centre for Environmental Research - UFZ, Brückstrasse 3a, D-39114 Magdeburg, Germany

[2] Département des Sciences Biologiques, Université du Québec à Montréal, Montréal, Québec, Canada

[3] Catalan Institute for Water Research (ICRA), Emili Grahit 101, 17003 Girona, Spain

[4] University of Girona, Girona, Spain

*Correspondence to*: Matthias Koschorreck ([matthias.koschorreck@ufz.de](mailto:matthias.koschorreck@ufz.de))

**Abstract.** Headspace analysis of $CO_2$ frequently has been used to quantify the concentration of $CO_2$ in freshwater. According to basic chemical theory, not considering chemical equilibration of the carbonate system in the sample vials will result in a systematic error. By analysing the potential error for different types of water and experimental conditions we show that the error incurred by headspace analysis of $CO_2$ is less than 5% for typical samples from boreal systems which have low alkalinity (<900 µmol $L^{-1}$), with pH (<7.5), and high $pCO_2$ (>1000 µatm). However, the simple headspace calculation can lead to high error (up to -300%) or even impossibly negative values in highly under saturated samples equilibrated with ambient air, unless the shift in carbonate equilibrium is explicitly considered. The precision of the method can be improved by lowering the headspace ratio and/or the equilibration temperature. We provide a convenient and direct method implemented in a R-script or a JMP add-in to correct $CO_2$ headspace results using separately measured alkalinity.

## 1. Introduction

The analysis of dissolved $CO_2$ in water is an important basis for the assessment of the role of surface waters in the global carbon cycle (Raymond et al., 2013). Indirect methods like calculating $CO_2$ from other parameters like alkalinity and pH (Lewis and Wallace, 1998; Robbins et al., 2010) are affected by considerable random and systematic errors (Golub et al., 2017) caused e.g. by dissolved organic carbon which may result in significant over estimation of the $CO_2$ partial pressure ($pCO_2$) (Abril et al., 2015), or by pH measurement errors (Liu et al., 2020). Thus, direct measurement of $CO_2$ is highly recommended, particularly in softwaters.

Headspace analysis is a standard method to analyse the concentration of dissolved gasses in liquids (Kampbell et al., 1989). In principle, a liquid sample is equilibrated with a gaseous headspace in a closed vessel under defined temperature. The partial pressure of the gas in the headspace is analysed, in most cases either by gas chromatography or infra-red spectroscopy. The concentration of the dissolved gas in solution is then calculated by applying Henry´s law after correction for the amount of gas transferred from the solution to the headspace.

In freshwater research this is the widely applied standard method to analyse the concentration of the greenhouse gases such as $CH_4$ and $N_2O$ (UNESCO/IHA, 2010). The method is handy, does not depend on sophisticated equipment in the field, and provides reliable results. Papers and protocols using this method have also been published to analyse dissolved $CO_2$ concentrations in freshwaters (UNESCO/IHA, 2010; Cawley, 2018; Lambert and Fréchette, 2005). However, $CO_2$ cannot be treated like $CH_4$ because $CO_2$ is in dynamic chemical equilibrium with other carbonate species in water while $CH_4$ is not (Stumm and Morgan, 1981; Sander, 1999). Depending on the $CO_2$ concentration and pH, reactions of the carbonate equilibrium will either produce or consume some $CO_2$ in the sample vessel (Cole and Prairie, 2009). Although this is textbook knowledge and has been considered in some recent papers (Golub et al., 2017; Gelbrecht et al., 1998; Rantakari et al., 2015; Aberg and Wallin, 2014; Horn et al., 2017), and is standard practice in marine research (Dickson et al., 2007), a practical evaluation of the systematic error when applying simple headspace analysis to $CO_2$ on typical freshwaters is missing, presumably because it is widely assumed that "the effect is likely small" (Hope et al., 1995). In this paper, we aim to quantify the error associated with the simple application of Henry´s law on headspace $CO_2$ data, present practical guidelines describing conditions under which the simple headspace analysis of $CO_2$ can give acceptable results, and offer a convenient tool for the exact $CO_2$ calculation that accounts for the carbonate equilibrium shifts in the sample equilibration vessel. The approach can also be used for correcting previous results obtained by simple headspace analysis of $CO_2$ using additional information regarding the carbonate system (i.e. alkalinity or DIC), a procedure we tested on a set of field measurements where $pCO_2$ was determined with independent methods (with and without headspace equilibration). Lastly, we evaluated how likely this correction may be required using a large data set from 337 diverse Canadian lakes.

## 2. Methods

### 2.1 Theoretical considerations

If a water sample is equilibrated with a headspace initially containing a known $pCO_2$ (zero in case $N_2$ or other $CO_2$-free gas is used), some $CO_2$ is exchanged between water and headspace resulting in an altered dissolved inorganic carbon (DIC) concentration in the water of the sample thereby altering the equilibrium of the carbonate system in the water. Depending on partial pressures of $CO_2$ in the water relative to the headspace gas prior to equilibration, some $CO_2$ will either be produced from $HCO_3^-$ or converted to $HCO_3^-$. The exact amount will depend on temperature, pH, total alkalinity (TA), and the original $pCO_2$ of the water sample. If a $CO_2$-free headspace gas was applied, the vessel will finally contain more $CO_2$ than before equilibration and consequently simply applying Henry´s law results in a too high $pCO_2$ value. If ambient air headspace is applied, the error becomes negative in under-saturated samples and the calculated $pCO_2$ an underestimate.

To calculate this error we implemented an R-script that simulates the above mentioned physical and chemical equilibration for a wide range of hypothetical $pCO_2$, alkalinity, temperature, and headspace ratio (HR = $V_{gas}$ / $V_{liquid}$) values. As output, we then compared the corrected (for the chemical equilibrium shift) and non-corrected $pCO_2$ values. All simulations were performed at 1 atm total pressure and results expressed as µatm.

### 2.2 Field data

As a further validation of our simulations, we used various data sets for which the $pCO_2$ was determined in multiple ways. We collated 266 observations from 4 reservoirs and 3 streams in Germany, 10 Canadian lakes, and a Malaysian reservoir exhibiting a wide range of TA between 0.03 and 1.9 mmol $L^{-1}$ and pH between 5.2 and 9.8. Two independent techniques were used to measure water $pCO_2$ in each sampling site: *in situ* NDIR technique and headspace equilibration technique. The same NDIR technique was used for all sites while the headspace technique differed slightly between sites. First, for the *in situ* NDIR technique, the water was pumped through the lumen side of a membrane contactor (mini module, Membrana, U.S.A.) (Cole and Prairie, 2009) and the gas side was connected to a NDIR analyser (EGM4, PP-Systems, U.S.A. or LGR ultra-portable gas analyser) in a counter-flow recirculating loop. Readings were taken when the $CO_2$ mole fraction (m$CO_2$ [ppm]) values of the NDIR analyser became stable (fluctuating ± 3 ppm around the mean) at which point the gas loop is in direct equilibrium with the sampled water. Final $pCO_2$ of the water was calculated by multiplying the m$CO_2$ by the ambient atmospheric pressure. Second, for the headspace technique, the methodology differed slightly among locations. In the German reservoirs, about 40 mL of water sample were taken in 60 mL syringes and eventually occurring bubbles were pushed out by adjusting the sample volume to 30 mL. Samples were stored at 4° C and analysed within 1 day. In the laboratory, 30 mL of pure $N_2$ gas was added to the syringes after the samples had reached laboratory temperature and the syringes were shaken for one hour at laboratory temperature. After headspace equilibration, the water was discarded from the syringes and the headspace was manually injected into a gas chromatograph equipped with a flame ionization detector (FID) and a methanizer (GC 6810C, SRI Instruments, U.S.A.). In the Canadian lakes, 20 mL of the water samples were taken in 60 mL syringes and equilibrated with 40 mL volume

of atmospheric air by vigorously shaking the syringes for 2 minutes. In the Malaysian reservoir, 600 mL of water samples were taken in 1.2 L of glass bottles and equilibrated with 611.5 mL of atmospheric air in 2016. In consecutive years, diverse volumes of water samples were taken in 60 mL or 100 mL syringes and equilibrated with diverse volumes of calibrated air brought from the laboratory. The equilibrated air was immediately transferred to and stored in 12 mL pre-evacuated exetainer vials (Labco Ltd., UK) and returned to the laboratory where it was injected into a gas chromatograph (GC-2014, Shimadzu, Kyoto, Japan) equipped with a FID. The original water $pCO_2$ was then calculated according to the headspace ratio, temperature, and the measured headspace $mCO_2$ as follows:

$$pCO_{2\ water} = \frac{(mCO_{2\ After\ eq} \times K_{h\ Eq} \times P) + \left\{ \left(\frac{V_{gas}}{V_{liquid}}\right) \times \left(\frac{mCO_{2\ After\ eq} - mCO_{2\ Before\ eq}}{V_m}\right) \right\}}{K_{h\ Sample}} \qquad \textbf{Eq. 1}$$

with $mCO_{2\ Before\ eq}$ and $mCO_{2\ After\ eq}$ are respectively the $CO_2$ mole fractions in the headspace before and after equilibrium [ppm], $K_{h\ Eq}$ and $K_{h\ Sample}$ = gas solubility at the equilibration temperature and at the sampling temperature (Henry coefficient (Sander, 2015)) [mol L$^{-1}$ atm$^{-1}$], P = pressure [atm], $V_{gas}$ = headspace volume, $V_{liquid}$ = sampled-water volume, and $V_m$ = molar volume [L mol$^{-1}$] (UNESCO/IHA, 2010). Results from Eq. 1 are reported as $pCO_2$ at one atmosphere of barometric pressure and are corrected for ambient pressure at the time of sampling by multiplying with the in situ atmospheric pressure.

The difference between headspace and NDIR method was divided by the $pCO_2$ measured by the *in situ* NDIR analysis and expressed as % error. In addition, temperature and pH of the water were measured *in situ* by a CTD probe (Sea and Sun, Germany) or a portable pH meter (pH meter 913, Metrohm Ltd, Canada). In samples from Canada and Germany, TA was analysed by titration with 0.11N HCl. In some systems, a single TA measurement was available for multiple dates and therefore assumes little temporal variability in the alkalinity of these systems. In the Malaysian samples, TA was derived from dissolved inorganic carbon (DIC) measurements and pH. Analysis of certified calibration gases showed that the analytical error of both the NDIR instrument and GC was <0.37% at 1000 ppm. Analysis of 7 replicate samples by our GC-headspace method gave a standard deviation of 6%. This includes all random errors due to sampling, sample handling and analysis.

To demonstrate the effect of our correction procedure, we used data from 377 lakes for which we had complete ancillary data and precise headspace measurements of $CO_2$ (<5% error between duplicates) obtained from the pan-Canadian Lake Pulse sampling program (Figure B1a, see Huot et al. (2019) for details).

### 3. Results and Discussion

**3.1 Simulations from chemical equilibrium**

Applying a $CO_2$-free gas as headspace always results in a positive error (over-estimation of the real $pCO_2$, Figure 1a). If ambient air is applied as headspace the error becomes negative in case of undersaturated samples (Figure 1b). In general, the error tends to be lower if ambient air is used for headspace equilibration (Figure 1b) compared to equilibration with $CO_2$-free gas (Figure 1a), except in undersaturated conditions. This is because less $CO_2$ is exchanged between water and headspace

during the equilibration procedure. The error will be below 5% in supersaturated and low alkalinity ($<900$ µmol L$^{-1}$) samples

which are typical for boreal regions. However, the error can be higher than 100% if the samples are undersaturated. The

magnitude of the error is predictable from pH. Because of the carbonate equilibrium reactions, high pH is necessarily

accompanied by low $pCO_2$ for a given alkalinity. Consequently, the error is large at high pH while it is below 10 % at pH < 8

(headspace gas:liquid ratio of 1:1).

Our field dataset is consistent with the theoretical predictions. While the fit between the simple headspace calculation and

NDIR values over the whole range of values can be considered adequate overall (Figure 2a, $R^2 = 0.92$), it is clear that the

deviations can become very large (up to about 300%), particularly at water $pCO_2$ values $<600$µatm (Figure 2b). As expected

from the simulations, the error in undersaturated samples was positive when using $CO_2$-free gas as headspace and negative

(sometimes impossible negative results) using ambient air (Figure 2b). The error became negligible at $pCO_2$ above 1000 µatm

(Figure 2b). Data scatter was considerable as was observed previously (Johnson et al., 2010), most probably because the

analytical error of the applied methods was often in the same range as the absolute difference between both methods.

### 3.2 Error magnitude depends on the experimental procedure

The maximum error depends on how much $CO_2$ is exchanged between water and headspace. The more gas is exchanged

between water and headspace the higher the error is. Thus, the error increases with decreasing solubility coefficient or HR. In

high alkalinity samples, the error can be significantly reduced by using a smaller headspace to water ratio (Figure 3). By

lowering the headspace ratio from 1 to 0.2 at 20°C the error can be reduced from about 50% to about 10%.

Since solubility of $CO_2$ depends on temperature, the equilibration temperature also affects headspace equilibration. Due to

lower solubility at higher temperature, more gas evades into the headspace and thus, the error increases with increasing

temperature (Figure 3a,b). At a HR of 1, the error increases from 97 % at 20°C to 111 % at 25°C in a high (1 mmol L$^{-1}$)

alkalinity sample. Thus, the error can be significantly reduced by lowering the equilibration temperature. A possible way to

take advantage of this is to perform headspace equilibration at *in situ* temperature in the field, as has been done in several

studies. If *in situ* water temperature is lower than typical laboratory temperature, the error is thereby reduced. However, care

must be taken to make sure that the exact equilibration temperature is known. For example, an error of 1°C in the equilibration

temperature results in a 2 % different $pCO_2$ value (TA=1 mmol L$^{-1}$, $pCO_2 = 1000$ µatm, HR = 1) (Figure A1a). Both ambient

air and N$_2$ can be used as headspace gas. Using N$_2$, however, eliminates the error associated with the exact quantification of

$pCO_2$ $_{Before}$. Using the same example, an unlikely error of 100 ppm in the headspace gas ($mCO_2$ $_{Before\ eq}$) results in a 6.4%

different $pCO_2$ result (Figure A1b).

### 3.3 What about kinetics?

$CO_2$ reactivity with water would not cause a problem for headspace analysis if the reaction kinetics were slow compared to

physical headspace equilibration. The slowest reaction of the carbonate system is the hydration of $CO_2$ which has a first order

rate constant of 0.037 s$^{-1}$ (Soli and Byrne, 2002) so that chemical equilibration of $CO_2$ in water is in the range of seconds
(Zeebe and Wolf-Gladrow, 2001; Schulz et al., 2006). This means that chemical equilibrium reactions are faster than physical
headspace equilibration and the chemical system can be assumed always to be in equilibrium. Thus, the reactions of the
carbonate system have to be fully considered in headspace analysis of $CO_2$.

**3.4 Correction of $CO_2$ headspace data**

If other information regarding the carbonate system of the sample is known (alkalinity or DIC), one can correct for the bias
induced by simple headspace calculations. A procedure to correct headspace $CO_2$ data using pH and alkalinity is already
available in the SOP N∘4 in Dickson et al. (2007) for marine samples and could be adapted to freshwater samples as well. For
convenience, we provide here a modified procedure when the alkalinity of the sample is known by introducing an analytical
solution to the equilibrium problem (iterative in SOP N∘4) and by using dissociation constants that may be more appropriate
to freshwaters. The procedure essentially involves estimating the exact pH of the equilibrium solution before and after
equilibration. If the alkalinity of the sample is known, the pH ($-\log_{10}[H^+]$ ) of the aqueous solution after equilibration can be
obtained by finding the roots of the 3$^{rd}$ order polynomial
$$0 = [H^+]^3 + TA \cdot [H^+]^2 - ([CO_2]K_1 + K_w)[H^+] - 2K_1K_2[CO_2]$$ **Eq. 2**
where $[CO_2] = pCO_2 \cdot K_{h\,Eq}$ and from which one can obtain the ionisation fraction for $CO_2$ ($\alpha_{CO2}$) as
$$\propto_{CO_2} = \frac{1}{1 + \frac{K_1}{[H^+]} + \frac{K_1K_2}{[H^+]^2}}$$ **Eq. 3**
where $K_1$ and $K_2$ are the temperature -dependent equilibrium constants for the dissociation reactions for bicarbonates and
carbonates, respectively (Millero, 1979), and for estuarine conditions, Millero (2010) as amended in Orr et al. (2015). $K_w$ is
the dissociation constant of water into $H^+$ and $OH^-$ (Dickson and Riley, 1979). The total DIC contained in the original sample
($DIC_{orig}$) can then be calculated as
$$DIC_{orig} = \frac{CO_2}{\alpha_{CO_2}} + (CO_{2\,HS_{after}} - CO_{2\,HS_{bef}})$$ **Eq. 4**
where $CO_2$ is the amount of $CO_2$ in the equilibrated water [mol], $CO_{2HS\,after\,+\,before}$ the amount of $CO_2$ in the headspace after
and before equilibration [mol]. Given the DIC concentration of the original solution from Eq. 4 ([DIC] = $DIC_{orig}$ / $V_{liquid}$), the
pH of this solution prior to equilibration can be obtained by finding the roots of the 4$^{th}$ order polynomial
$$0 = [H^+]^4 + (TA + K_1) \cdot [H^+]^3 + (TA \cdot K_1 - K_w + K_1K_2 - [DIC]_{orig}K_1) \cdot [H^+]^2 + (K_1K_2 \cdot TA - K_1K_w - 2[DIC]_{orig}K_1K_2) \cdot [H^+] -$$
$$K_1K_2K_w$$ **Eq. 5**
to then estimate the corresponding ionization fraction $\alpha'_{CO_2}$ as in Eq. 3 above and calculate the original $pCO_2$ of the sample
as
$$pCO_2 = \frac{\alpha'_{CO_2} \cdot [DIC]_{orig}}{K_{h\,Sample}}$$ **Eq. 6**
where $K_{h\,Sample}$ is determined for the water temperature during field sample collection (for simplicity, the equations above
assume a 1 atm pressure). We applied the above correction procedure to our samples where pCO$_2$ was measured in several
samples using both headspace and in situ NDIR methods together with measured alkalinity data. Figure 4 shows that the
corrected values matched the in situ NDIR values nearly perfectly ($r^2$=0.98) whereas the simple headspace calculations
resulted, as expected, in significant underestimation for undersaturated conditions, particularly for samples equilibrated with
ambient air.
We examined the sensitivity of the correction procedure to the precision of the alkalinity measurements and found that the
error associated with alkalinity determination does not severely impact the final pCO$_2$ estimate when using N$_2$ as a headspace
gas. For example, the error in the corrected pCO$_2$ values is always below 20% even when the alkalinity is known only to within
50% error (Fig. 3c). However, more precise alkalinity values are required when using ambient air as a headspace gas in
undersaturated conditions (Fig. 3d).
Lastly, our simulations (Figs. 2 and 4) provide a complete analysis of the effects of the environmental and methodological
conditions on the error incurred when using the simple headspace technique for estimating pCO$_2$. However, they do not assess
how often such problematic conditions occur in inland water systems. To address this question, we applied our correction
procedure to a dataset from 377 Canadian lakes (Huot et al., 2019). These results show a significant deviation between
corrected and uncorrected values, particularly in lakes with high alkalinity (>900 $\mu$mol L$^{-1}$, Figure B1b) and ignoring the
correction would have resulted errors >20% in about 47% of the data. Furthermore, our analysis illustrates how a larger
headspace ratio significantly exacerbates the magnitude of the error (Figure B1b).
The correction calculations have been implemented in an R script and, for a user-friendly interface, as an JMP add-in (or JSL
script) (https://github.com/icra/headspace). Roots of the polynomials (Eqs. 2 and 5) can be solved using either standard
analytical formulas or by iterative algorithms. For the analytical solution, our script uses a combined form of the computational
steps described in Zwillinger (2018) for both the cubic and quartic polynomials to find their first real roots. Analytical solutions
are faster than iterative algorithms but can suffer small numerical instabilities (SD ≈1 ppm) in extreme situations (alkalinity
>4000$\mu$mol L$^{-1}$ and pCO$_2$<100ppm) due to limitations inherent to double precision numerical calculations. The provided scripts
consider the barometric pressure and thus, allow calculation of pCO$_2$ as well as CO$_2$ concentration [$\mu$mol L$^{-1}$] for *in situ*
conditions.
**4. Conclusions**
The headspace method has been used in several studies about CO$_2$ fluxes from surface waters. Our error analysis shows that
the usual headspace method can be used (error<5%) if the pH is below 7.5 or pCO$_2$ is above 1000 $\mu$atm (TA< 900 $\mu$mol L$^{-1}$,
air headspace), a typical situation in most boreal systems. However, the standard headspace method introduces large errors
and cannot be used reliably for under saturated samples, which are typical of eutrophic or low DOC systems. In all other cases,
not accounting for the chemical equilibrium shift leads to a systematic over estimation. The magnitude of the error can be
reduced by increasing the water/headspace ratio or lowering the equilibration temperature. The magnitude of that error can be
roughly estimated from Figure 1. If alkalinity is known, $pCO_2$ obtained from headspace equilibration can be corrected by the
provided scripts. We therefore recommend to always measure alkalinity if the headspace method is to be used for $pCO_2$
determinations. The procedure can also be used to correct historical $pCO_2$ data. Our field data showed that the correction works
well even in highly undersaturated conditions and is not very sensitive to the precise determination of alkalinity if $N_2$ is used
as a headspace gas. The precision of the corrected $pCO_2$ is similar to that obtained from direct $pCO_2$ measurement using a field
NDIR analyser coupled to an on-line equilibrator (Cole and Prairie, 2009; Yoon et al., 2016).

## 5.  Appendices

**Appendix A: Sensitivity analysis equilibration temperature and $CO_2$ Before eq**

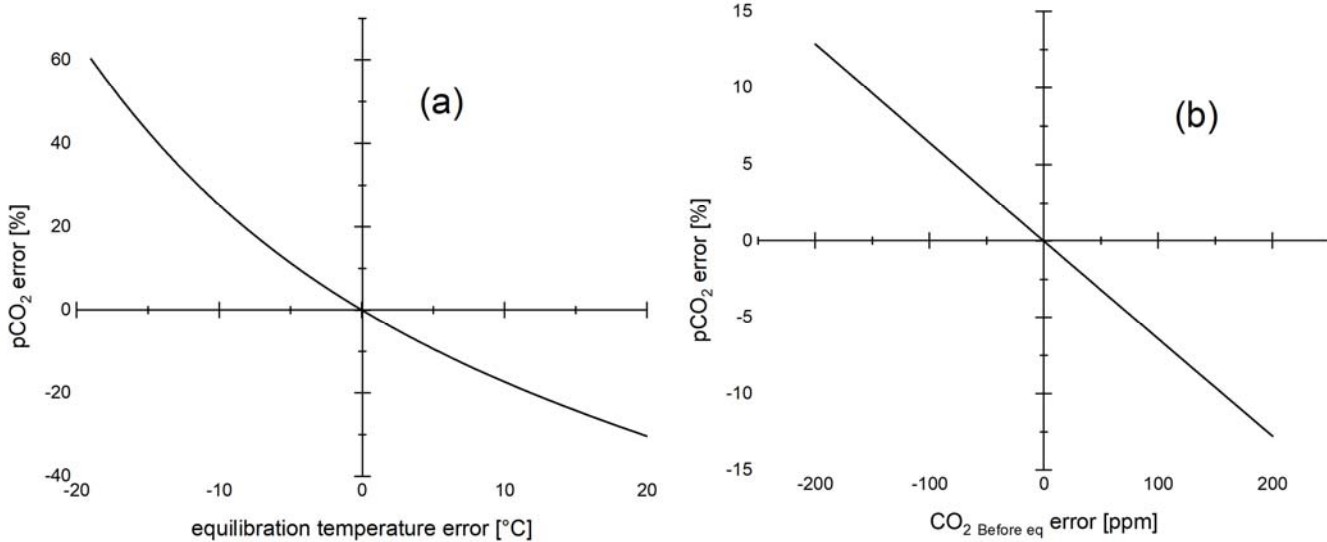

**Figure A1: Error for a hypothetical sample with $CO_2$ Before eq = 400 ppm, $CO_2$ after eq = 1000 ppm, equilibration**
**temperature 20°C, HR = 1 (a) depending on error in equilibration temperature (b) depending on error in initial**
**headspace gas composition.**

 **Appendix B: Application of our correction to a large Canadian dataset**

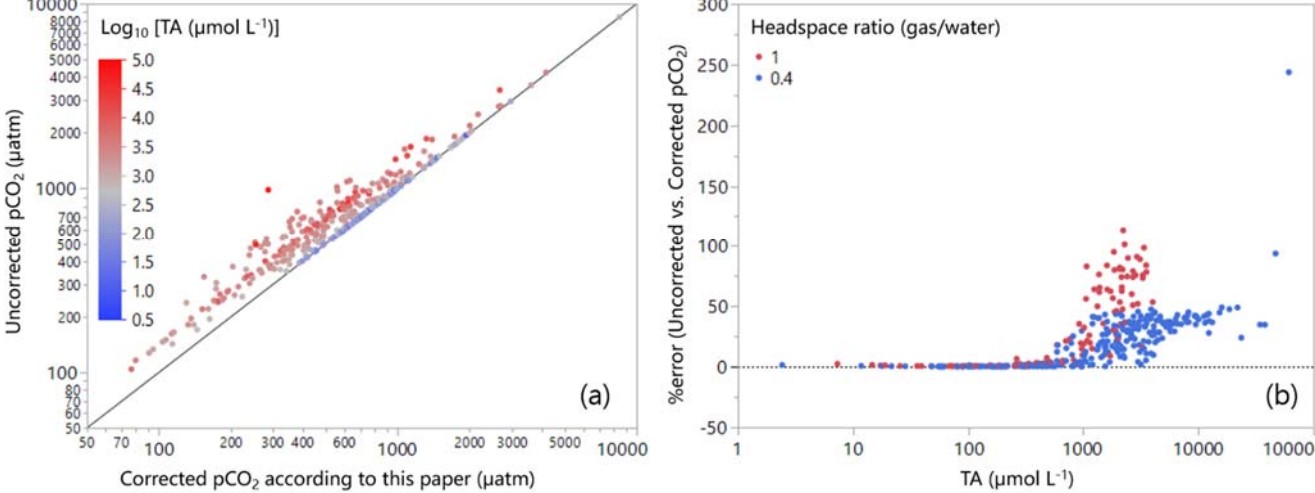


**Figure B1: Field data from 377 lakes across Canada (a) for comparing pCO₂ derived from simple headspace calculation**
**with that from the corrected headspace calculation according to this paper (Log₁₀ [TA (µmol L⁻¹)] colour coded). (b)**
**Difference between the uncorrected and corrected pCO₂ expressed as error (%) as a function of TA (µmol L⁻¹) (The**
**headspace ratio colour coded). Note that CO₂-free gas was used for headspace, and TA values were derived from DIC**
**measurement and pH. More information about the dataset in Huot et al. (2019).**
**6. Code availability**
All codes are publicly available at https://github.com/icra/headspace
**7. Data availability**
All data can be found in the supplemental information file.
**8. Supplement link (will be included by Copernicus)**
**9. Author contribution**
All authors conceived the story, performed calculations, and wrote the manuscript. JHK, YP, and RM wrote codes. MK and
JHK contributed field data.

## 10. Competing interests

The authors declare that they have no conflict of interest.

## 11. Acknowledgements

Thanks to Philipp Keller, Peifang Leng, Lelaina Teichert, and Cynthia Soued for sharing additional field data     and Alo Laas for logistic support. Thanks to Miitta Rantakari, Marcus Wallin, and Pascal Bodmer for stimulating discussions and to Bertram Boehrer for commenting on the manuscript. We also thank MT Trentman, JR Blaszczak, and RO Hall Jr for an independent check of our calculations during the public discussion of the manuscript. RM participated through the project C-HYDROCHANGE, funded by the Spanish Agencia Estatal de Investigación (AEI) and Fondo Europeo de Desarrollo Regional (FEDER) under the contract FEDER-MCIU-AEI/CGL2017-86788-C3-2-P. RM acknowledges the support from the Economy and Knowledge Department of the Catalan Government through Consolidated Research Group (ICRA-ENV 2017 SGR 1124), as well as from the CERCA program. JHK and YTP were funded by the Natural Sciences and Engineering Research Council of Canada. This is a contribution to the UNESCO Chair in Global Environmental Change.

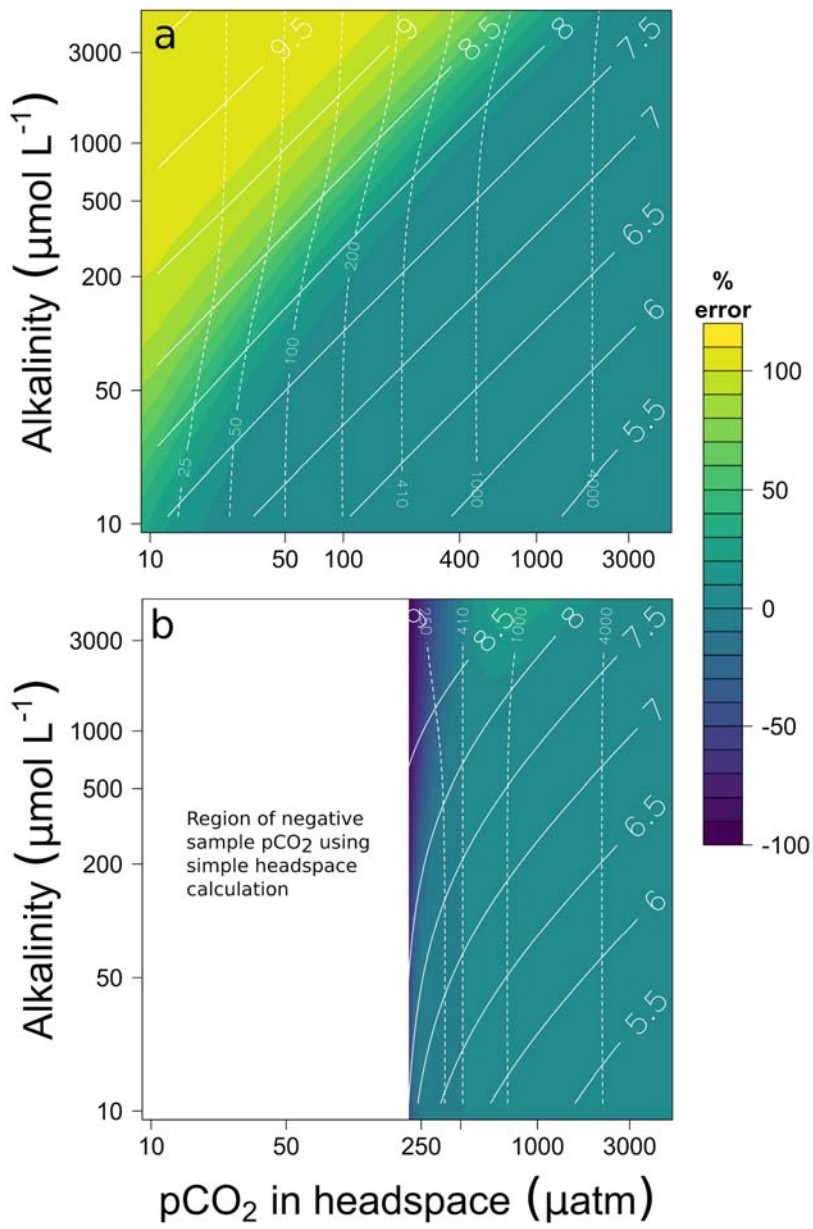

**Figure 1: Error [%] when applying simple headspace calculations of pCO₂ on hypothetical water samples of different alkalinity and**
**pCO₂ in the headspace after equilibration for (a) CO₂-free gas headspace and (b) ambient-air headspace assuming a pressure of 1**
**atm. The resulting pH and pCO₂ of the samples are depicted as full and dashed lines, respectively. Headspace ratio 1:1, equilibration**
**and field temperature 20°C. Note the log scale in all axes. In b) results for pCO₂ in headspace after equilibration lower than 215**
**µatm are masked, because they would imply negative pCO₂ in the sample.**

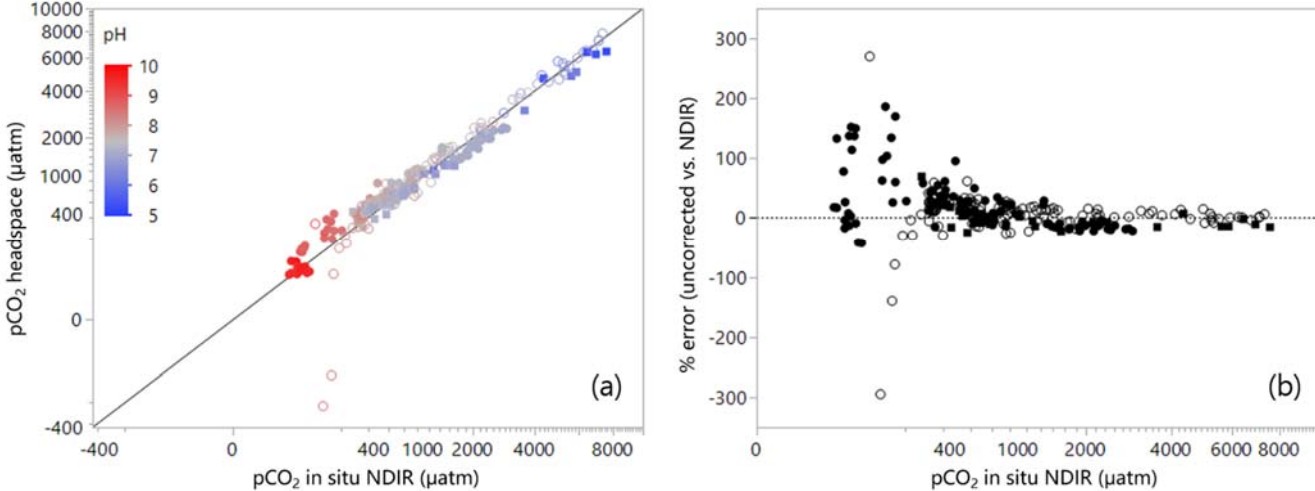


**Figure 2: (a) Field data from 11 lakes, 5 reservoirs, and 3 streams in Germany, Canada, and Malaysia comparing pCO₂ derived**
**from simple headspace analysis with direct pCO₂ measurements by NDIR analysis (pH colour coded). Note the cube-root scale in**
**both axes. (b) Difference between the pCO₂ derived from the simple headspace analysis and the direct pCO₂ measurements by NDIR**
**analysis expressed as error (%) as a function of the directly measured pCO₂ by NDIR analysis. Note the cube-root scale in x axis.**
**Open-circle symbols: ambient-air headspace, closed-circle symbols: CO₂-free gas headspace, and closed-square symbols:**
**premeasured-CO₂ gas (between 150 to 250 ppm) headspace applied.**


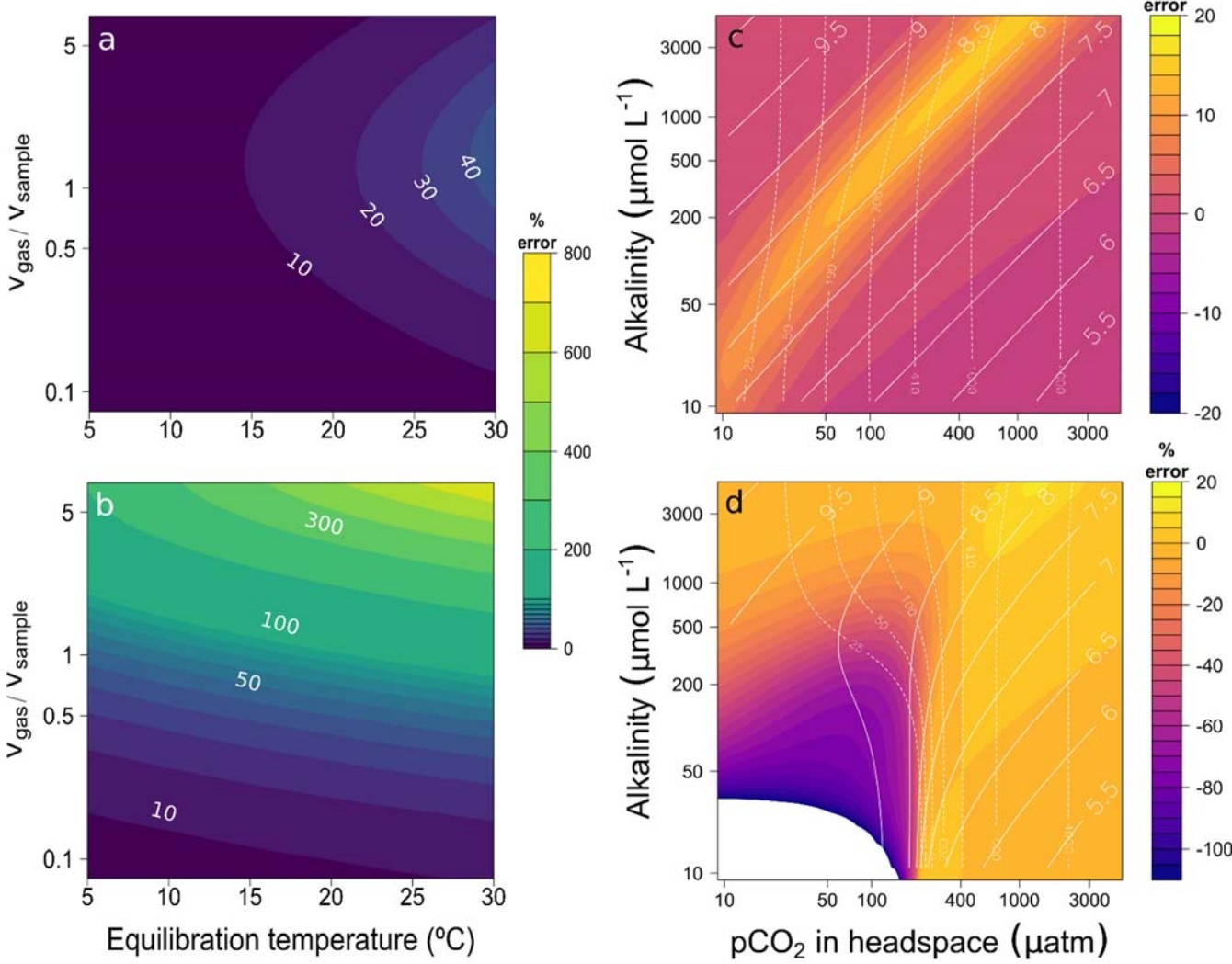


Figure 3: Error [%] when applying simple headspace calculation depending on headspace ratio and equilibration temperature for a) 100 µmol L$^{-1}$ and b) 1000 µmol L$^{-1}$ alkalinity. Panels a and b were constructed using highly undersaturated conditions (headspace pCO$_2$=50 µatm after equilibration and field water temperature of 20ºC). The values of some isolines are added for reference. c) Error [%] applying our complete headspace method when the alkalinity value supplied for calculations is off the real alkalinity of the sample by +50%. The results are for hypothetical water samples of different alkalinity and pCO$_2$ in the headspace after equilibration using CO$_2$-free gas headspace, headspace ratio 1:1, and equilibration and field temperature of 20°C. d) like c) but with air headspace. All calculations assume a pressure of 1 atm.

280

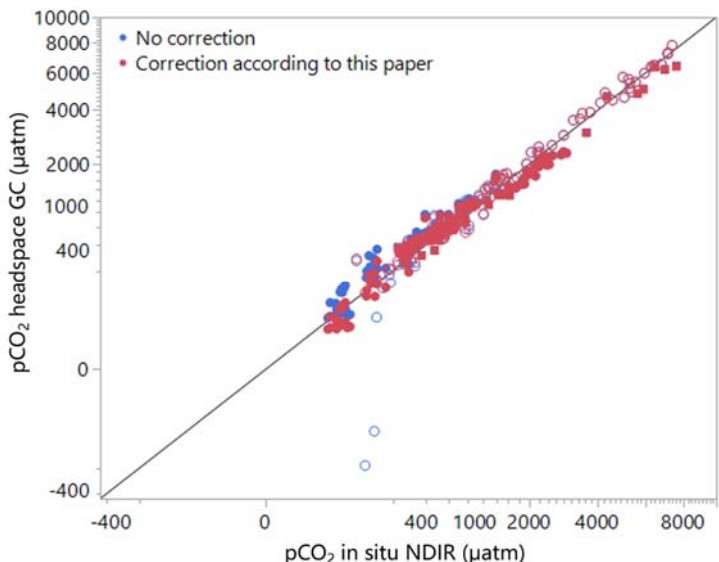

**Figure 4: Comparison of uncorrected and corrected data with direct pCO₂ measurements by NDIR analysis. Note the cube-root scale in both axes. Open-circle symbols: ambient-air headspace, closed-circle symbols: CO₂-free gas headspace, and closed-square symbols: premeasured-CO₂ gas (between 150 to 250 ppm) headspace applied.**

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
