# Peer review of "Technical note: CO2 is not like CH4 – limits of and corrections to the headspace method to analyse pCO2 in freshwater"

_Biogeosciences, 2020_

## Referee Comment (RC1) · Anonymous Referee #1 · 3 Sep 2020

The importance of this paper is to point out a major problem in the processing of CO2 measurements made by headspace equilibration due to the equilibration of CO2 with HCO3- to the wider community working on CO2 dynamics in freshwater.

This has been known for decades by the marine CO2 community, related to the buffering capacity of water due to the presence of HCO3- that in fact strongly affects all aspects of CO2 dynamics in marine and freshwater environments.

This problem was possibly less acknowledged by the freshwater CO2 community due to the dominance of soft-water lakes in northern North America and Scandinavia where the very large majority of studies of inland water CO2 studies have been carried out so

far.

That said, the authors reinvent the wheel by proposing a "tool for exact CO2 calculation" because the marine CO2 community has established for decades a method to correct the CO2 data from measurements of headspace. This is the SOP N°4 ("Determination of pCO2 in air that is in equilibrium with a discrete sample of sea") of the two versions of the "CO2 Handbook" (DOE 1994; Dickson et al. 2007).

This method can be also applied to the type of data reported by the authors by computing DIC from TA and pCO2_After_eq, correcting DIC for the CO2 loss or gain during equilibration in the headspace (based on pCO2_After_eq and pCO2_Before_eq and using the law pf perfect gases), and re-computing "correct" pCO2 (pCO2_water) from TA and corrected DIC.

The SOP4 method also allows to correct for water temperature changes between in-situ water and water sample after equilibration. This change of temperature can be substantial (depending on the difference between air temperature and in-situ water temperature) and will lead to a strong bias of pCO2 values. For the first step of the computation of DIC the water temperature of sample after equilibration is used. For the final step of the computation of pCO2 (from corrected DIC) the in-situ temperature is used giving corrected pCO2 at in-situ temperature.

I suggest that the authors should mention SOP4 in the ms and compare both "tools".

Finally, I find it regretful that the authors did not reach out to the community for additional data-sets that would have made their case more compelling by extending the range of pCO2 and Total alkalinity values, and thus more representative of lakes globally. Several groups have obtained similar data-sets of direct pCO2 measurements by equilibration coupled to NDIR detectors in parallel with pCO2 measurements based headspace equilibration, and could have been contacted.

References

Dickson, A.G., Sabine, C.L. and Christian, J.R. (Eds.) 2007. Guide to best practices for ocean CO2 measurements. PICES Special Publication 3, 191 pp.

DOE. 1994. Handbook of methods for the analysis of the various parameters of the carbon dioxide system in sea water; version 2, A.G. Dickson and C. Goyet, Eds. ORNL/CDIAC-74

---

## Author Comment (AC1) · 4 Sep 2020

We thank the reviewer for the quick assessment of our work and the constructive comments it includes.

The reviewer is correct that this topic is well-known in marine science. However, for multiple reasons, the potential impact of the carbonate equilibrium on headspace CO2 calculation never percolated to a good portion of the freshwater community. There are many examples in the freshwater literature where pCO2 was determined from headspace equilibration without any consideration of the altered chemical equilibrium during the equilibration process. Perhaps this should be more clearly stated. As the

reviewer points out, one of the main reason is that most of the freshwater literature deals with very soft waters and many scientists intuitively recognize the problem but simply consider its effect to be small and therefore neglect it. So, our impetus was not to "re-invent the wheel" but simply to draw attention that the issue is not always negligible, even in soft waters. This is the reason we wrote this manuscript as a small technical note in a journal widely read by the freshwater community.

The reviewer is also quite right to point out that we should have made more direct references to the marine literature and its SOPs (although the marine SOP4 was cited already, Dickson et al. 2007). This will be further amended in the revision process. Nevertheless, there are some differences between the SOP4 of the marine community and what we propose. For example, the equations used for temperature-dependence of the equilibrium constants in marine SOPs are not the most suitable for freshwater samples (even when Sal is set to zero) and our code provides alternative equation sets for different environments (freshwater, estuarine, marine). Also, while the procedure in Annex 2 of SOP4 reproduces the same logic, the solution we provide does not require an iterative solution-finding algorithm but are instead exact solutions and therefore do not require an initial H+ estimate. In freshwaters, pH is much more variable than in the ocean (pH of lakes typically between 5 and 9.5). Nevertheless, we also provide an iterative procedure. We will modify the paper accordingly to clarify these subtle differences and will include a comparison between our tool and the SOP4 protocol as suggested by the reviewer.

While it is true that SOP4 (with some modifications to account for the low salinity typical of many inland waters) would provide a viable alternative to calculate CO2 using the headspace method also in freshwater, the fact remains that its use within the freshwater community is rare and considered largely anecdotal despite its existence for decades. We feel that this is reason enough to alert the freshwater community of its potential significance and more importantly, identify when and where it is particularly relevant. Our intention was not to claim that we have developed something completely

novel, but to draw attention of this issue to the freshwater community, and offer a tool specifically targeted to freshwater researcher. In our opinion, an easy to use tool may boost the adoption of the same procedural principles adopted in marine sciences but that are not used by the freshwater community. Further, the codes we provide to do corrections in an easy way should encourage its wider use. This makes our manuscript especially suited as a "technical report" and will hopefully help to improve data quality in freshwater carbon research.

As for the effect of temperature changes during equilibration, we will revise our wording in the manuscript because in fact this is considered in our calculations. We will make sure that this is clear in a revised version of the manuscript.

We will consider the reviewer's comment on the necessity of contacting other groups to obtain more coupled NDIR and headspace data but felt that the pool of our own relevant data was sufficient to exemplify our point.

We are happy to keep discussing these or other aspects of our manuscript.

Regards,

Matthias Koschorreck, Yves T. Prairie, Jihyeon Kim, and Rafael Marcé

---

## Referee Comment (RC2) · Anonymous Referee #2 · 18 Sep 2020

The authors describe a method to correct pCO2 measurements using the headspace method with discrete water samples for the effect of the CO2 equilibrium in fresh water samples. They compare different approaches of the headspace technique with measurements using a membrane equilibrator directly in the field. I mostly agree with the comments from Reviewer #1 that most of the work is described in Dickson et al (2007) and I will not repeat the arguments here. I understand that the authors want to use this technical note to raise awareness in the community. I can't judge if it is worth to publish an article repeating knowledge that is well documented (as the authors state themselves), or if there are better ways to raise the awareness. But I assume a technical note might be the right way. The authors talk a lot about errors in %. As an outsider

of the fresh water community, I was missing a short introduction into the field given the aimed accuracies/precisions of pCO2 measurements. Also the analytical errors of their methods are not stated. This makes it hard to evaluate the benefit of any correction. An error of 100% means to me that the measurement is not very useful, but I might be wrong here. As reviewer #1 said, the manuscript would benefit of comparing the two correction methods, the one described here with the one presented in Dickson et al. (2007). More specific comments: l. 41: a space is missing between UNESCO/IHA, 2010; and Cawley...) l. 58: dissolved l. 61: alkalinity (TA); TA is already introduced here, so it can be skipped in line 69 l.74: What is stable? Can you give the range? l. 87: Is the equation correct? The units do not cancel out to a pressure. l. 88: pCO2 should be given in pressure units not in ppm l. 92: What do you mean with the "two methods"? NDIR vs. headspace method? l. 101: Please give a range and not only "quite". l. 107: what is acceptable? l. 116ff and Fig. 3: I don't get the text and the Figure together. Do you mean "using a smaller headspace"? When I look at Fig. 3b at 20°C and HR=1 and move to HR=5 the error increases to 300%.

Dickson, A.G., Sabine, C.L. and Christian, J.R. (Eds.) 2007. Guide to best practices for ocean CO2 measurements. PICES Special Publication 3, 191 pp.

---

## Short Comment (SC1) · 11 Oct 2020

The technical note quantifies the potential sampling error in estimates of dissolved CO2 concentrations from headspace measurements, which is caused by shifts in the carbonate equilibrium in the sample. I have been using the headspace method for estimating pCO2 in inland waters for many years. Despite being aware of the general problem, particularly at high pH, the associated error has rarely been considered or even quantified. The technical note provides an excellent analysis of the potential errors for a range of relevant sampling conditions in freshwater ecosystems. I consider the presented results as an easy way for checking the expected error for both planning

of sampling campaigns and for possible correction of the measurements. As pointed out by both reviewers, the limitations of the headspace technique for measuring pCO2 are more considered in the marine science community. I acknowledge the contributions that this technical note makes for increasing the awareness of this problem in the freshwater community and by providing quantitative error assessments and correction methods.

---

## Author Comment (AC2) · 13 Oct 2020

Similar to reviewer one, this review questions whether our technical note is necessary given that it is already acknowledged in the marine sciences. Our view is that there is actually a very strong need for it in the freshwater community considering the large number of studies not considering the potential artifact of simple CO2 headspace calculation. A good example is a recently published paper about errors in pCO2 calculation which uses the NEON dataset (which is based on simple headspace calculation) as reference (https://aslopubs.onlinelibrary.wiley.com/doi/full/10.1002/lom3.10388). The question becomes really what is the best way to raise awareness of this overlooked

problem in our segment of the scientific community. For this purpose we opted to write a short technical note in which we not only identify the problem but also 1) perform a quantitative assessment of the potential error (which to our knowledge never have been published in a systematic way before) and point out the general circumstances where this is most problematic, and 2) provide an easy to use tool to correct both old and future data. To date, the positive feedback we received from a number of colleagues confirms our belief that this note is a valuable contribution and therefore should improve the quality of future freshwater carbon cycle studies. To make this clearer we consider exchanging the "water" in the title by "freshwater". Besides this general point the reviewer raises a couple of minor issues which we would all address in a revised manuscript. We will correct the naming of variables in Eq. 1 by renaming pCO2Aftereq and pCO2Beforeeq to mrCO2Aftereq and mrCO2Beforeeq because these numbers are, as we explain in the text mixing ratios and not partial pressures. We will further add quantitative information about analytical errors and our NDIR method and will fix the other minor edits identified by the reviewer.

―――――――――――――――――――――

---

## Short Comment (SC2) · 15 Oct 2020

**Comment: Freshwater $CO_2$ Headspace Equilibration Calculations**

M. T. Trentman[1], J. R. Blaszczak[2], R. O. Hall, Jr.[1]

[1]Flathead Lake Biological Station, University of Montana, Polson, MT USA

[2]Department of Natural Resources and Environmental Science,

University of Nevada-Reno, Reno, NV USA

October 15, 2020

Koschorreck et al. describe a method for calculating the original $CO_2$ concentration of a water sample when using a headspace equilibration to extract dissolved $CO_2$ from water. This manuscript contributes a correction to the final measured $CO_2$ concentration to account for the near-instantaneous equilibration between $CO_2$ and $HCO_3^-$ in water during equilibration with the headspace. While the $HCO_3^-$ equilibration correction exists in the chemical oceanography literature (e.g., Dickson et al. 2007), this correction is used less frequently in the freshwater literature. We have multiple ongoing projects in freshwater streams involving data using headspace equilibrations for estimating $CO_2$ concentrations, and until reading this work, we had not considered correcting $CO_2$ concentrations for the $HCO_3^-$ equilibrium despite always having a copy of Dickson et al. (2007) nearby. Thus, we believe this manuscript is a welcome addition to the literature. Given the importance of accurate calculations for estimating $CO_2$ concentrations and our previous experience with these calculations, our objectives in this comment are to:

1. Use a different mathematical formulation than in Koschorreck et al. to estimate the $HCO_3^-$ equilibrium correction, and compare the magnitude of correction between the two approaches.

2. Evaluate the correction magnitude with samples where headspace equilibration may bias estimates of $pCO_2$: high alkalinity and a large volume $CO_2$-free headspace relative to the water volume.

We used our own grab sample data to compare the full set of headspace equilibration calculations as presented by Koschorreck et al. with the same calculations derived from our group. We note that in parts of our calculations we use different equations than Koschorreck et al. but the approaches are based on the same theory. Our code and a detailed comparison of our calculations with that of Koschorreck et al. are available at https://github.com/jrblaszczak/CO2_headspace_code. First, we added the $HCO_3^-$ equilibrium correction

to our existing headspace equilibration calculations using equations 4 and 5 in Dickson et al. (2007) (SOP 4). We then compared the calculated $CO_2$ concentration from the code provided by Koschorreck et al. with our code using diel grab sample data (sampled every 2-4 hr over a 24 hr period) from streams in northwestern Montana and central Arizona. For these samples, we performed headspace equilibrations in the field using 40 mL of streamwater and 70 mL of injected $CO_2$-free air in 140 mL syringes. We equilibrated the headspace by shaking the syringes for 3 minutes, after which we flushed the water from the syringe and stored the remaining gas sample in the syringe until analysis within 48 hours on a Picarro G2131-i analyzer (Picarro, Santa Clara, CA, USA). We measured stream temperature and barometric pressure at the time of water sampling and collected an unfiltered streamwater sample for total alkalinity, which was measured by titration to a pH of 4.5. Stream temperature of our samples was between 7 and 22 °C and the total alkalinity was between 1440 and 2060 $\mu$eq $L^{-1}$.

There was minimal variation of the $HCO_3^-$ equilibrium corrections and corrected $CO_2$ concentrations between the calculations provided by Koschorreck et al. and the calculations we independently compiled. The $HCO_3^-$ equilibrium corrections between the approaches were similar to the hundredth decimal place (expressed as $\mu$mol $L^{-1}$ DIC), which we consider to be functionally the same. The average percent deviation of corrected $CO_2$ concentrations between approaches was 0.9% (SD $\pm$ 0.6%) which is likewise a small difference. As noted above, the code we provide for this comparison uses different calculations than Koschorreck et al., but is based on the same theory. Thus, we see this comparison as an independent verification of their calculations.

The deviation of $HCO_3^-$ equilibrium corrected and uncorrected $CO_2$ concentrations for our samples ranged between (2-23%, reported in ppmv). The error in our samples is not surprising given the relatively high total alkalinity, low headspace ratio (1.75), and that we used a $CO_2$-free headspace. We are limited in our ability to change our methods to reduce this error given our protocol for measuring gases on the Picarro G2131-i analyzer requires 70 mL of gas per sample, and using $CO_2$-free headspace is logistically easier than adding a measurement of air for each sample during a diel. Additionally, equilibration air containing $CO_2$ would bias estimates of $\delta^{13}$C-DIC. By adding the $HCO_3^-$ equilibrium correction we can increase the accuracy of our grab sample $CO_2$ concentrations without altering our sampling protocol.

While our approaches result in the same outcome we note one difference that is worth mentioning by Koschorreck et al.. This difference involves assumptions we make about calculating the $H^+$ concentration (eqn. 2, Koschorreck et al.) from the equation for total alkalinity ($A_T$, eqn. 1) that leads to an algebraically simpler equation and therefore may enable more efficient computation and incorporation of these calculations into larger process models.

$$A_T = [HCO_3^-] + 2[CO_3^{-2}] + [OH^-] - [H^+] \tag{1}$$

We assume that the $H^+$ and $OH^-$ concentrations are zero in eqn. 1 for two reasons. First, these concentrations are tiny compared to $HCO_3^-$ and $CO_3^{-2}$, and likely within the range of the error associated with the measured alkalinity via titration or charge balance. Thus, we contend that assuming $H^+$ and $OH^-$ concentrations are zero will have a negligible effect on the calculation of the overall $H^+$ concentration. Second, this assumption allows us to solve a $2^{nd}$-order polynomial rather than a $3^{rd}$-order polynomial equation. While the authors incorporate an elegant solution to solve the $3^{rd}$-order polynomial using the *polyroot* function in R, the use of this function may limit downstream incorporation of these calculations in stochastic simulations. For example, we use adaptations of Markov chain Monte Carlo methods (e.g., the general purpose Bayesian modeling software Stan, Carpenter et al. 2017) to simulate posterior distributions in Bayesian hierarchical models to estimate stream metabolism from diel patterns of $CO_2$. In this case, the simpler mathematical solution for solving for the $H^+$ concentration will facilitate computational speed without sacrificing accuracy. Given the occasional need for mathematically simpler solutions, we suggest the authors discuss the complexity of their approach, and note that the calculations can be simplified by assuming $[H^+]$ and $[OH^-] \approx 0$.

Overall, we feel that Koschorreck et al. provide an useful contribution to the literature that will lead to more accurate measurements of $CO_2$ concentrations, particularly in freshwaters. The extensive analyses by Koschorreck et al. of the deviation of corrected and uncorrected $CO_2$ concentrations across geochemical and methodological scenarios provide necessary context to this issue. Likewise, Koschorreck et al. are the first group to our knowledge to combine calculations for $CO_2$ concentrations from headspace equilibrations into a streamlined and publicly available R script. We commend the authors for pointing out a commonly neglected correction and for providing the code with which the community can easily overcome this additional step to estimating accurate dissolved $CO_2$ concentrations.

We have a few minor comments on the manuscript:

1. We suggest that the authors carefully re-examine and edit the text in the abstract and conclusion sections that highlights the general importance of factors that may impact the deviation between corrected and uncorrected $CO_2$ concentrations for consistency. For example, in the conclusion it is noted that samples with pH below 7.5 and $pCO_2$ above 1000 $\mu$atm will have a small error, but in the abstract only the pH is noted. We also note that the abstract and conclusion sections differ in the suggested content of gas used in the headspace to reduce error. The abstract states $CO_2$ free gas should be used while the conclusion states that air should be used instead of an $N_2$ headspace. Consistency of the messaging of these factors would provide more clarity to the manuscript.

2. We suggest the authors use more consistent notations, particularly for $pCO_2$. The authors could note both the 'location' of the $pCO_2$ measurement (i.e., headspace vs. water) as well as before or

after equilibrium in their notation. This caused some confusion for us when evaluating the text and code. A few examples include: line 141- $pCO_2$ is used while in line 87 $pCO_{2water}$ is used, line 87- $pCO_{2Aftereq}$ is used while in line 146 $pCO_{2HSafter}$ is used. We believe consistent annotation will improve understanding of these calculations.

3. Many researchers, including our group, prefer to report our data in units of $\mu$mol $L^{-1}$, especially when we compare data of different molecules (e.g., $CO_2$ vs $O_2$ for metabolism). Thus, we suggest that the authors include in the R code the reporting of $CO_2$ concentrations in $\mu$mol $L^{-1}$. This should not be much trouble given the calculations are already conducted in molar units.

**References**

Carpenter, B., Gelman, A., Hoffman, M. D., Lee, D., Goodrich, B., Betancourt, M., Brubaker, M., Guo, J., Li, P., and Riddell, A.: Stan: A probabilistic programming language, Journal of Statistical Software, 76, 1–32, doi:10.18637/jss.v076.i01, 2017.

Dickson, A. G., Sabine, C. L., Christian, J. R., and Bargeron, C. P., eds.: Guide to best practices for ocean CO2 measurements, no. 3 in PICES special publication, North Pacific Marine Science Organization, Sidney, BC, 2007.

Koschorreck, M., Prairie, Y. T., Kim, J., and Marcé, R.: Technical note: CO2 is not like CH4; limits of and corrections to the headspace method to analyse pCO2 in water, Biogeosciences, pp. 1–12, URL https://bg.copernicus.org/preprints/bg-2020-307/, 2020.

---

## Author Comment (AC3) · 6 Nov 2020

We thank Mark Trentman and his colleagues for their detailed and constructive comment to our manuscript. We appreciate the independent verification of our calculations – an exercise which was actually asked for by the first reviewer. We also acknowledge their frank recognition that they, like many others, did not consider carbonate speciation in their headspace analysis in the past. This strongly supports our argument that this is a useful contribution to the freshwater community.

Trentman et al. recommend to add a statement that a simpler way of calculation could be used under the assumption that H+ and OH- concentrations are negligible. This

would, as they claim, save computing power and would facilitate integration of the correction into models. We are not really convinced about the necessity of this simplification, given current computing power and our analytical solution can easily be integrated in any process-based models. Why should one use a less precise method in the presence of our solution? Furthermore, the assumption of negligible H+ and OH- might be true for high pH and alkalinity waters, but doesn't necessarily hold under low alkalinity conditions typically found in boreal environments where, for example, [OH-] can easily be of the same magnitude as [CO32-]. We see the strength of our approach in the fact that it can equally be applied to all types of freshwaters.

Besides this, Trentman et al raise some minor issues which we will address in a revision: We will propose clear recommendations which headspace gas should be used under which conditions. We will also clarify notations. We will also include calculation of molar concentrations in our script (already included in the JMP version).

---

## Author Response (AR1)

The importance of this paper is to point out a major problem in the processing of $CO_2$ measurements made by headspace equilibration due to the equilibration of CO2 withHCO3- to the wider community working on $CO_2$ dynamics in freshwater. This has been known for decades by the marine CO2 community, related to the buffering capacity of water due to the presence of HCO3- that in fact strongly affects all aspects of $CO_2$ dynamics in marine and freshwater environments. This problem was possibly less acknowledged by the freshwater CO2 community due to the dominance of soft-water lakes in northern North America and Scandinavia where the very large majority of studies of inland water $CO_2$ studies have been carried out so far. That said, the authors reinvent the wheel by proposing a "tool for exact CO2 calculation" because the marine CO2 community has established for decades a method to correct the CO2 data from measurements of headspace. This is the SOP N∘4 ("Determination of pCO2 in air that is in equilibrium with a discrete sample of sea") of the two versions of the "CO2 Handbook" (DOE 1994; Dickson et al. 2007).This method can be also applied to the type of data reported by the authors by computing DIC from TA and pCO2_After_eq, correcting DIC for the CO2 loss or gain during equilibration in the headspace (based on pCO2_After_eq and pCO2_Before_eq and using the law pf perfect gases), and re-computing "correct" pCO2 (pCO2_water) from TA and corrected DIC.

The SOP4 method also allows to correct for water temperature changes between in-situ water and water sample after equilibration. This change of temperature can be substantial (depending on the difference between air temperature and in-situ water temperature) and will lead to a strong bias of pCO2 values. For the first step of the computation of DIC the water temperature of sample after equilibration is used. For the final step of the computation of pCO2 (from corrected DIC) the in-situ temperature is used giving corrected pCO2 at in-situ temperature. I suggest that the authors should mention SOP4 in the ms and compare both "tools".

As stated in the introduction, we are aware that carbon speciation is routinely considered in the marine literature and have cited the widely used SOP no.4 in our manuscript. We have highlighted this in the revised manuscript with the sentence "A procedure to correct headspace $CO_2$ data using pH and alkalinity is already available in the SOP N∘4 in Dickson et. al. (2007) for marine samples and could be adapted to freshwater samples as well. For convenience, we provide here a modified procedure when the alkalinity of the sample is known by introducing an analytical solution to the equilibrium problem to facilitate calculations (iterative in SOP no. 4) and by using dissociation constants that may be more appropriate to freshwaters." We directly compared our correction with SOP no4 and got very similar values when the whole range of samples was considered. We feel such comparison actually tackles the more difficult question of whether freshwaters can be assimilated to very dilute seawater and therefore justify the use of dissociation constant formulas that were

determined by sequential dilutions of seawater. For example, when we compared the SOP no.4 method with our procedure on the system where accounting for the shifting equilibrium was most needed (highly undersaturated system using ambient air as headspace), we found that the NDIR values were closer to our estimation than to the SOP no.4, essentially because of the difference in dissociation constants used (see figure). While the question of the most adequate dissociation constants for freshwaters is definitely relevant, it is considerably beyond the

scope of this technical note. Our main purpose here is instead to alert the freshwater community of the importance of accounting for the shift in chemical equilibrium during headspace equilibration and under what circumstances it is most problematic. However, the reviewer's comments have prompted us to better articulate this main message and we have now included a completely new section directly addressing the question of how relevant it is to the limnological community. To this end, we have compared the results of $pCO_2$ determinations with and without correction from a large dataset of 377 lakes from across Canada for which we had complete ancillary data and precise headspace measurements of $CO_2$ (<5% error between duplicates). These results show that ignoring the correction would have resulted errors >20% in about 50% of the data. This new paragraph is now towards the end of section 3.4.

Regarding the effect of equilibration temperature vs in situ temperature, it was already included in our procedure and also considered in all our data but we had failed to mention it. After Equ 1 we write "$K_{h\,Eq}$ and $K_{h\,Sample}$ = gas solubility at the equilibration temperature and at the sampling temperature". We also added a figure to the appendix which shows the sensitivity of the results towards errors in the equilibration temperature and address the importance of using the correct temperature by writing "However, care must be taken to make sure that the exact equilibration temperature is known. For example, an error of 1°C in the equilibration temperature results in a 2 % different pCO2 value (TA=1 mmol L-1, pCO2 = 1000 µatm, HR = 1) (Figure A1a)".

Finally, I find it regretful that the authors did not reach out to the community for additional data-sets that would have made their case more compelling by extending the range of pCO2 and Total alkalinity values, and thus more representative of lakes glob-ally. Several groups have obtained similar data-sets of direct pCO2 measurements by equilibration coupled to NDIR detectors in parallel with pCO2 measurements based headspace equilibration, and could have been contacted.

The reviewer is right and we agree that Figure 4 can benefit from more data expanding the geographical coverage and range of water chemistry. To this end, we contacted a number of colleagues and have added data from 3 reservoirs and 3 streams in Germany as well as from a Malaysian reservoir. Thus, our dataset now contains 266 observations a variety of systems types in different continents covering a large range in pH and alkalinity. We added to the method section: "We sampled water in 4 reservoirs and 3 streams in Germany, 10 Canadian lakes, and a Malaysian reservoir exhibiting a wide range of TA between 0.03 and 2.4 meq L$^{-1}$ and pH between 5.2 and 9.8.". As written above we also applied our correction to a large dataset of 377 lakes from across Canada to demonstrate the need for proper correction.

**References**

Dickson, A.G., Sabine, C.L. and Christian, J.R. (Eds.) 2007. Guide to best practices for ocean CO2 measurements. PICES Special Publication 3, 191 pp.DOE. 1994. Handbook of methods for the analysis of the various parameters of the carbon dioxide system in sea water; version 2, A.G. Dickson and C. Goyet, Eds.ORNL/CDIAC-74

**Anonymous Referee #2**

The authors describe a method to correct pCO2 measurements using the headspace method with discrete water samples for the effect of the CO2 equilibrium in freshwater samples. They compare different approaches of the headspace technique with measurements using a membrane equilibrator directly in the field. I mostly agree with the comments from Reviewer #1 that most of the work is described in Dickson et al (2007) and I will not repeat the arguments here. I understand that the authors want to use this technical note to raise awareness in the community. I can't judge if it is worth to publish an article repeating knowledge that is well documented (as the authors state

themselves), or if there are better ways to raise the awareness. But I assume a technical note might be the right way. The authors talk a lot about errors in %. As an outsider of the fresh water community, I was missing a short introduction into the field given the aimed accuracies/precisions of pCO2 measurements. Also the analytical errors of their methods are not stated. This makes it hard to evaluate the benefit of any correction. An error of 100% means to me that the measurement is not very useful, but I might be wrong here.

We added information about analytical errors at the end of the method section: "Analysis of certified calibration gases showed that the analytical error of both the NDIR instrument and GC was <0.37% at 1000 ppm. Analysis or 7 replicate samples by our GC-headspace method gave a standard deviation of 6%. This includes all random errors due to sampling, sample handling and analysis." We now also assess the error resulting from wrong equilibration temperature by writing "For example an error of 1°C in the equilibration temperature results in a 2% different $pCO_2$ value (TA=1 meq $L^{-1}$, $pCO_2$=1000 µatm, HR = 1).".

As reviewer #1 said, the manuscript would benefit of comparing the two correction methods, the one described here with the one presented in Dickson et al.(2007).

We have already addressed the comparison between SOP4 in our response to reviewer #1.

More specific comments:

l. 41: a space is missing between UNESCO/IHA,2010; and Cawley...)

corrected

l. 58: dissolved

corrected

l. 61: alkalinity (TA); TA is already introduced here, so it can be skipped in line 69

corrected

l.74: What is stable? Can you give the range?

We added "fluctuating ± 5 ppm around the mean".

l.87: Is the equation correct? The units do not cancel out to a pressure.

Yes - the equation is correct. Confusion with the units arose because of the misleading notation of $pCO_2$ which in fact is a dimensionless mole fraction. This is corrected in the equation and text.

l.88: pCO2should be given in pressure units not in ppm

Since gas chromatograph results are typically reported as mixing ratios we would like to keep ppm here. Instead, we renamed $pCO_2$ to $mCO_2$ as explained above.

l.92: What do you mean with the "two methods"? NDIR vs. headspace method?

Yes. We changed the sentence to "The difference between headspace and NDIR method…".

l.101: Please give a range and not only "quite".

We specified to "below 10%".

l.107: what is acceptable?

Good question. What is acceptable depends on the precision wanted and thus can be different for different studies. We removed the "acceptable" and replaced it by a more detailed description of the error: "While the fit between the simple headspace calculation and NDIR values over the whole range of values can be considered adequate overall (Figure 2a, $R^2$ = 0.92), it is clear that the deviations can become very large (up to about 300%), particularly at water $pCO_2$ values <600µatm (Figure 2b). As expected from the simulations, the error in undersaturated samples is positive when using $CO_2$-free gas as headspace and negative (sometimes impossibly so) using ambient air (Figure 2b)."

l.116ff and Fig. 3: I don't get the text and the Figure together. Do you mean "using a smaller headspace"? When I look at Fig. 3b at20∘C and HR=1 and move to HR=5 the error increases to 300%.

Correct and thank you for pointing out this error. As written in the manuscript any measure reducing the gas exchange between water and headspace should reduce the error – and with a smaller headspace less gas is exchanged. We changed the sentence to: "In high alkalinity samples, the error can be significantly reduced by using a smaller headspace to water ratio (Figure 3). By lowering the headspace ratio from 1 to 0.25 at 20° the error can be reduced from about 50% to about 10%."

**reference**

Dickson, A.G., Sabine, C.L. and Christian, J.R. (Eds.) 2007. Guide to best practices for ocean CO2 measurements. PICES Special Publication 3, 191 pp

**Comment: Freshwater CO2 Headspace Equilibration Calculations**

M. T. Trentman, J. R. Blaszczak, R. O. Hall, Jr.

Koschorreck et al. describe a method for calculating the original CO2 concentration of a water sample when using a headspace equilibration to extract dissolved CO2 from water. This manuscript contributes a correction to the final measured CO2 concentration to account for the near-instantaneous equilibration between CO2 and HCO3 in water during equilibration with the headspace. While the HCO3 equilibration correction exists in the chemical oceanography literature (e.g., Dickson et al. 2007), this correction is used less frequently in the freshwater literature. We have multiple ongoing projects in freshwater streams involving data using headspace equilibrations for estimating CO2 concentrations, and until reading this work, we had not considered correcting CO2 concentrations for the HCO□

3 equilibrium despite always having a copy of Dickson et al. (2007) nearby. Thus, we believe this manuscript is a welcome addition to the literature. Given the importance of accurate calculations for estimating CO2 concentrations and our previous experience with these calculations, our objectives in this comment are to:

1. Use a different mathematical formulation than in Koschorreck et al. to estimate the HCO3 equilibrium correction, and compare the magnitude of correction between the two approaches.
2. Evaluate the correction magnitude with samples where headspace equilibration may bias estimates ofpCO2: high alkalinity and a large volume CO2-free headspace relative to the water volume.

We used our own grab sample data to compare the full set of headspace equilibration calculations as presented by Koschorreck et al. with the same calculations derived from our group. We note that in parts of our calculations we use different equations than Koschorreck et al. but the approaches are based on the same theory. Our code and a detailed comparison of our calculations with that of Koschorreck et al. are available at https://github.com/jrblaszczak/CO2 headspace code. First, we added the HCO3 equilibrium correction to our existing headspace equilibration calculations using equations 4 and 5 in Dickson et al. (2007) (SOP 4). We then compared the calculated CO2 concentration from the code provided by Koschorreck et al. with our code using diel grab sample data (sampled every 2-4 hr over a 24 hr period) from streams in northwestern Montana and central Arizona. For these samples, we performed headspace equilibrations in the field using 40 mL of streamwater and 70 mL of injected CO2-free air in 140 mL syringes. We equilibrated the headspace by shaking the syringes for 3 minutes, after which we flushed the water from the syringe and stored the remaining gas sample in the syringe until analysis within 48 hours on a Picarro G2131-i analyzer (Picarro, Santa Clara, CA, USA). We measured stream temperature and barometric pressure at the time of water sampling and collected an unfiltered streamwater sample for total alkalinity, which was measured by titration to a pH of 4.5. Stream temperature of our samples was between 7 and 22 _C and the total alkalinity was between 1440 and 2060 µeq L$^{-1}$.

There was minimal variation of the $HCO_3$ equilibrium corrections and corrected $CO_2$ concentrations between the calculations provided by Koschorreck et al. and the calculations we independently compiled. The $HCO_3$ equilibrium corrections between the approaches were similar to the hundredth decimal place (expressed as µmol $L^{-1}$ DIC), which we consider to be functionally the same. The average percent deviation of corrected $CO_2$ concentrations between approaches was 0.9% (SD = 0.6%) which is likewise a small difference. As noted above, the code we provide for this comparison uses different calculations than Koschorreck et al., but is based on the same theory. Thus, we see this comparison as an independent verification of their calculations.

Thank you for this independent check of our procedure

The deviation of $HCO_3$ equilibrium corrected and uncorrected $CO_2$ concentrations for our samples ranged between (2-23%, reported in ppmv). The error in our samples is not surprising given the relatively high total alkalinity, low headspace ratio (1.75), and that we used a $CO_2$-free headspace. We are limited in our ability to change our methods to reduce this error given our protocol for measuring gases on the Picarro G2131-I analyzer requires 70 mL of gas per sample, and using $CO_2$-free headspace is logistically easier than adding a measurement of air for each sample during a diel. Additionally, equilibration air containing $CO_2$ would bias estimates of $_{13}C$-DIC. By adding the $HCO_3^-$ equilibrium correction we can increase the accuracy of our grab sample $CO_2$ concentrations without altering our sampling protocol.

While our approaches result in the same outcome we note one difference that is worth mentioning by Koschorreck et al.. This difference involves assumptions we make about calculating the $H^+$ concentration (eqn. 2, Koschorreck et al.) from the equation for total alkalinity ($A_T$, eqn. 1) that leads to an algebraically simpler equation and therefore may enable more efficient computation and incorporation of these calculations into larger process models.

$$A_T = [HCO_3^-] + 2[CO_3^{-2}] + [OH^-] - [H^+] \tag{1}$$

We assume that the $H^+$ and OH- concentrations are zero in eqn. 1 for two reasons. First, these concentrations are tiny compared to $HCO_3^-$ and $CO_3^{-2}$ and likely within the range of the error associated with the measured alkalinity via titration or charge balance. Thus, we contend that assuming $H^+$ and $OH^-$ concentrations are zero will have a negligible effect on the calculation of the overall $H^+$ concentration. Second, this assumption allows us to solve a 2nd-order polynomial rather than a 3rd-order polynomial equation. While the authors incorporate an elegant solution to solve the 3rd-order polynomial using the polyroot function in R, the use of this function may limit downstream incorporation of these calculations in stochastic simulations. For example, we use adaptations of Markov chain Monte Carlo methods (e.g., the general purpose Bayesian modeling software Stan, Carpenter et al. 2017) to simulate posterior distributions in Bayesian hierarchical models to estimate stream metabolism from diel patterns of $CO_2$. In this case, the simpler mathematical solution for solving for the $H_+$ concentration will facilitate computational speed without sacrificing accuracy.

Given the occasional need for mathematically simpler solutions, we suggest the authors discuss the complexity of their approach, and note that the calculations can be simplified by assuming $[H^+]$ and $[OH^-] \sim 0$.

As we already commented in our short reply during the discussion phase, the advantages of using an approximation are not clear given current computing power and, more importantly, given that our analytical solution can easily be integrated in any process-based stochastic models (more easily than an iterative solution would be). Our script takes 0.00086 s to run a sample, or 14 minutes for one million runs. It is unlikely that these running times limit the applicability of our method. Also, while the assumption of negligible $H^+$ and $OH^-$ might be true for high pH and alkalinity waters, it doesn't necessarily hold under low alkalinity conditions typically found in boreal environments where, for example, $[OH^-]$ can easily be of the same magnitude as $[CO_3^{2-}]$. We see the strength of our approach in the fact that it can equally be applied to all types of freshwaters.

Overall, we feel that Koschorreck et al. provide a useful contribution to the literature that will lead to more accurate measurements of $CO_2$ concentrations, particularly in freshwaters. The extensive analyses by Koschorreck et al. of the deviation of corrected and uncorrected $CO_2$ concentrations across geochemical and methodological scenarios provide necessary context to this issue. Likewise, Koschorreck et al. are the first group to our knowledge to combine calculations for $CO_2$ concentrations from headspace equilibrations into a streamlined and publicly available R script. We commend the authors for pointing out a commonly neglected correction and for providing the code with which the community can easily overcome this additional step to estimating accurate dissolved $CO_2$ concentrations.

We acknowledge this point in the abstract by modifying the last sentence to "We provide a convenient direct method implemented in a R-script or a JMP add-in to correct $CO_2$ headspace results using separately measured alkalinity.

We have a few minor comments on the manuscript:
1. We suggest that the authors carefully re-examine and edit the text in the abstract and conclusion sections that highlights the general importance of factors that may impact the deviation between corrected and uncorrected $CO_2$ concentrations for consistency. For example, in the conclusion it is noted that samples with pH below 7.5 and $pCO_2$ above 1000 µatm will have a small error, but in the abstract only the pH is noted. We also note that the abstract and conclusion sections differ in the suggested content of gas used in the headspace to reduce error. The

abstract states $CO_2$ free gas should be used while the conclusion states that air should be used instead of an $N_2$ headspace. Consistency of the messaging of these factors would provide more clarity to the manuscript. We apologize for these inaccuracies. We removed the statement about which headspace gas should be used from the abstract since the potential error is similar for both. We also specified the conditions for low error in the abstract by writing "By analysing the potential error for different types of water and experimental conditions we show that the error incurred by headspace analysis of $CO_2$ is less than 5% for typical samples from boreal systems which have low alkalinity (<1700 µmol $L^{-1}$), low pH (with pH <7.5), and high $pCO_2$ (>1000 µatm).

2. We suggest the authors use more consistent notations, particularly for $pCO_2$. The authors could note both the `location' of the $pCO_2$ measurement (i.e., headspace vs. water) as well as before or after equilibrium in their notation. This caused some confusion for us when evaluating the text and code. A few examples include: line 141- $pCO_2$ is used while in line 87 $pCO_{2water}$ is used, line 87- $pCO_{2Aftereq}$ is used while in line 146 $pCO_{2HSafter}$ is used. We believe consistent annotation will improve understanding of these calculations.

We improved the notations as recommended by the reviewer. Notably we now use "$mCO_2$" for molar mixing ratios and "$pCO_2$" for partial pressure.

3.Many researchers, including our group, prefer to report our data in units of µmol $L^{-1}$, especially when we compare data of different molecules (e.g., $CO_2$ vs $O_2$ for metabolism). Thus, we suggest that the authors include in the R code the reporting of $CO_2$ concentrations in µmol $L^{-1}$. This should not be much trouble given the calculations are already conducted in molar units.

We agree. The JMP script and add-in already provided corrected values in both partial pressure and molar unit. The R script now provides the same.

**References**

Carpenter, B., Gelman, A., Ho_man, M. D., Lee, D., Goodrich, B., Betancourt, M., Brubaker, M., Guo, J., Li, P., and Riddell, A.: Stan: A probabilistic programming language, Journal of Statistical Software, 76, 1{32, doi:10.18637/jss.v076.i01, 2017.

Dickson, A. G., Sabine, C. L., Christian, J. R., and Bargeron, C. P., eds.: Guide to best practices for ocean CO2 measurements, no. 3 in PICES special publication, North Paci_c Marine Science Organization, Sidney, BC, 2007.

Koschorreck, M., Prairie, Y. T., Kim, J., and Marcé, R.: Technical note: CO2 is not like CH4; limits of and corrections to the headspace method to analyse pCO2 in water, Biogeosciences, pp. 1{12, URL https://bg.copernicus.org/preprints/bg-2020-307/, 2020.